# Linguistic Explanations of Black Box Deep Learning Detectors on Simulated Aerial Drone Imagery

**DOI:** 10.3390/s23156879

**Published:** 2023-08-03

**Authors:** Brendan Alvey, Derek Anderson, James Keller, Andrew Buck

**Affiliations:** Department of Electrical Engineering and Computer Science, University of Missouri, Columbia, MO 65211, USA; andersondt@missouri.edu (D.A.); kellerj@missouri.edu (J.K.); buckar@missouri.edu (A.B.)

**Keywords:** aggregation, black box, deep learning, evaluation, Explainable AI, fuzzy, linguistic, object detection, simulation

## Abstract

Deep learning has become increasingly common in aerial imagery analysis. As its use continues to grow, it is crucial that we understand and can explain its behavior. One eXplainable AI (XAI) approach is to generate linguistic summarizations of data and/or models. However, the number of summaries can increase significantly with the number of data attributes, posing a challenge. Herein, we proposed a hierarchical approach for generating and evaluating linguistic statements of black box deep learning models. Our approach scores and ranks statements according to user-specified criteria. A systematic process was outlined for the evaluation of an object detector on a low altitude aerial drone. A deep learning model trained on real imagery was evaluated on a photorealistic simulated dataset with known ground truth across different contexts. The effectiveness and versatility of our approach was demonstrated by showing tailored linguistic summaries for different user types. Ultimately, this process is an efficient human-centric way of identifying successes, shortcomings, and biases in data and deep learning models.

## 1. Introduction

A trained object detection model may perform well under circumstances that are similar to its training and validation data. However, current generation artificial intelligence (AI) and machine learning (ML) do not yet behave well in new and novel scenarios that deviate from training. Identifying the limitations and capabilities of a black-box AI/ML model is an important, timely, relevant, and challenging task. Our aim was two-fold. First, we desired a human-centric process for extracting tailored linguistic statements for eXplainable AI (XAI) and domain knowledge discovery. Second, we desired a human-free solution that leverages recent advancements in text/natural language-based generative AI and procedural simulation. The objective was a closed loop process for automatic model search and refinement. Figure 1 presents an overview of these two complementary yet competing goals.

Recent developments in large language models (LLMs) have led to groundbreaking new tools. LLMs are AI models trained on vast amounts of text data which output human-like text responses. Many of the highest performing models use extensions of the Transformer neural network architecture [1]. Although language models have existed for quite some time, the relatively recent drastic increase in scale, along with techniques for zero-shot learning and reinforcement learning from human feedback have resulted in unprecedented breakthroughs [2,3,4]. Modern LLMs have been shown to preform exceedingly well in a number of tasks including translation, writing essays, creating poetry, code generation, and summarization [5,6,7,8,9]. While the output from an LLM is far from perfect, it is sometimes preferred over human generated responses [10]. In the current article, we demonstrate a hybrid approach which combines more traditional machine generated linguistic explanations of an object detector with an LLM to yield an improved user-friendly report. Although LLMs are being researched for understanding and explaining neural networks themselves [11], these methods are not yet mature enough to be trusted in many contexts. Often, LLMs can “hallucinate” poor responses [12] that are difficult to explain and result in reduced trust in a system. It is also not currently clear how exactly LLM provides explanations and, hence, whether they should be trusted. As a direct result, we opted for a more transparent and explainable approach that seeks to reduce the weaknesses associated with both approaches. Linguistic summarization is fully explainable but it results in a relatively complete set of explanations in a somewhat artificial language format. On the other hand, an LLM cannot yet compute all explanation tasks nor do we entirely trust the explanations it yields. Herein, linguistic summarization produces a set of user tailored explanations, and an LLM can reduce these explanations into a succinct natural language description.

LLMs can also be used in combination with other techniques to generate novel photorealistic imagery from relatively simple, natural language text prompts [13,14]. Language models can be integrated with sophisticated rendering systems, such as the Unreal Engine, to create diverse photorealistic imagery with a simple set of instructions [15,16]. In this paper, we demonstrated an XAI system for generating textual descriptions of an object detection model. A future goal is to directly use the output from our system when tuned to focus on failure modes to generate novel imagery for training. This forms a closed loop learning system, capable of identifying shortcomings and addressing them to the best of its ability.

Many datasets, e.g., for unmanned aerial vehicles (UAV)/drones, are becoming larger in size and more detailed in terms of attributes. By recording camera, environmental, and target parameters associated with each image, we can automatically conduct a more detailed analysis of model performance than we could with only images and object locations. As the number of data attributes increases, so too does the number of possible descriptions and ways of analyzing the data. Herein, we explored a new and novel process to automatically generate a summarized set of linguistic statements of a black box AI/ML model evaluated on data with many underlying data attributes. The objective was to produce a reduced and tailored set of target explanations surrounding successes, failures, and possible data and model biases. To demonstrate the proposed ideas, an AI/ML model that was trained on real-world data for person detection was evaluated on simulated photorealistic aerial data produced by the Unreal Engine. By using a virtual drone and the Movie Render Queue feature in Unreal Engine, we generated a simulated person dataset. An advantage of simulation is it has access to dense and accurate truth. Figure 2 shows an example scene with a person, a simulated drone, and an image that is captured from the low altitude aerial drone’s point of view.

Linguistic summaries are a family of approaches which automatically create human-interpretable descriptions of datasets. They have been used to detect falls, improve the treatment of elders, forecast time series data, and in many other applications [17,18,19,20,21,22,23,24].

One of the largest challenges in effectively analyzing AI/ML model behavior has been simply obtaining the data. It is very expensive and time consuming to collect and annotate dense and accurate real-world aerial imagery. In many applications, there is a training and/or evaluation bottleneck. Thankfully, due to tools like the Unreal Engine, Unity [25], Apple’s Hypersim [26], and NVIDIA’s Isaac Gym [27], we can now quickly produce large and diverse sets of photorealistic imagery with dense and accurate truth. While these tools are not a perfect simulation or a replacement for the real-world, their level of realism is high and multiple works are already showing how they can be used as is to train and evaluate AI/ML algorithms [28,29,30,31,32,33]. In [34], we showed how this imagery can be used to systematically evaluate model performance. Through simulation, we are able to output very detailed descriptions for each image in our data set. However, this is initially a double edged sword with respect to linguistic summaries. For each new data attribute (e.g., camera elevation angle, solar azimuth angle, altitude, etc.), the number of possible statements describing the data increases exponentially. It is important to note that, while simulation is a large focus of the proposed article, it is not a necessity. If the reader can collect truth in the real-world, that information can be used instead. We primarily focused on simulation because it is a controlled environment within which o study the proposed methods, and exploring its possible role in AI/ML is an open research question.

Let a linguistic statement, *S*, be a “Linguistic Protoform Summary” of the type “Q R Ys are P”, where Q is a quantifier, R is a qualifier, Ys are data samples and P is a summarizer. A quantifier designates the amount of samples supported by the qualifier and summarizer. One attribute is reserved as the qualifier. It can be a performance metric such as the intersection over union (IOU) or confidence, combined with a fuzzy predicate, $F_{R}$, (e.g., low, medium, high). Summarizers are constructed from the power set of attributes and their fuzzy-predicates. Let $Y=\{y_{1},y_{2},\cdots ,y_{N}\}$ be a dataset of *N* samples. Objects are represented as a set of attribute–value pairs. As mentioned above, one attribute is reserved as the qualifier while the other *K* attributes are used to construct the summarizer. An example attribute–value pair for an image would be [“altitude”, 10 m″]. Each attribute, *k*, has a list of $F_{k}$ fuzzy predicates associated with it. The value of the *k*th attribute for the *n*th data sample is denoted as $A_{k}\left(y_{n}\right)$. Each attribute and fuzzy-predicate pair has a fuzzy membership function associated with it, $\mu_{k,j}$. This function maps the value that corresponds to each attribute to a fuzzy membership value between 0 and 1. An example of such functions is shown in Figure 3. Use Table 1 as a reference for the notation used in this paper.

The number of possible statements as a factor of the number of quantifiers, $J_{Q}$, and fuzzy-predicates is $J_{Q}J_{R}\prod_{k=1}^{K}\left(J_{k}+1\right)$. This means moving from three quantifiers and three fuzzy-predicates for the qualifier and three other attributes to four quantifiers with four fuzzy-predicates for the qualifier and five other attributes is an increase from 576 to 50,000 possible statements. Making sense of the huge number of possible descriptions for a system is a major topic of this paper. We factor in linguistic qualities such as truth, focus, complexity, and a user-specified filter we call operational relevancy.

Expressing the state and behavior of a system using natural human language is the goal of linguistic summaries. Yager first introduced an approach to generating linguistic summaries for relational databases using the theory of fuzzy subsets [35]. Statements were created of the form, “Q objects in Y are P”, where Q is known as a “Quantity in Agreement” such as “Few” or “Many”. Y is the dataset being summarized, and P is a description such as “Altitude” or “Camera Elevation Angle”. Fundamental to this approach is the use of fuzzy set theory. For each possible statement, Yager computed a “Truth” value, τ. This truth value is computed by first calculating the degree to which each sample satisfies each summarizer. That value is then passed through the fuzzy membership function for the quantity in agreement of the statement. The “Truth” value is just one of many criteria we can consider when trying to analyze linguistic summaries. Often, we may prefer a few short statements that can efficiently communicate important information, rather than many lengthy, highly specific statements. The complexity of a statement can be measured by either counting the number of linguistic components, or simply counting the number of words or characters in a complete statement to determine its length.

Kacprzyk [36] proposed using five “quality” measures for filtering linguistic summaries—truth, imprecision, covering, appropriateness, and length. Imprecision is a value computed based on how vague or precise each statement is. For example, “On some of the days its somewhat hot” is not as precise as “On every day it is cold”. Coverage is a measure of how much of a dataset is actually represented by a particular statement. A statement such as “High confidence detections at high altitudes” may happen to have a high truth measure but be poorly supported by few samples of data. If we use more than one criteria for filtering linguistic summaries, then we must also consider how to best aggregate them together. It is not a trivial task choosing which combination is best; there are many options [37]. One may use a pessimistic intersection like operator (t-norm) or an optimistic union like operator (t-conorm). An approach proposed by Popek and Katarzyniak is to use an interval-based aggregation [38]. Their method aims to build an internal representation of linguistic concepts to relate different statements. For this application, we chose to use a different approach, described in Section 2.3.

In summary, the current article was focused on generating tailored linguistic statements of black box AI/ML model performance. A simulation was used to build datasets with truth relative to object detection on a low altitude aerial drone. The following sections detail these steps. In Section 2, we outline the structure and process for extracting summaries. In Section 3, experiments and results are provided. Last, we summarize our findings and discuss future work.

## 2. Theoretical Background

In this section, we formally define our linguistic summary process. Let *Y*={$y_{1}$, $y_{2}$, ⋯, $y_{N}$} be a dataset of *N* samples. Each sample in our dataset is an image that comes with associated metadata: a list of K+1 key-value data attributes that describe the sample. (These attributes are either produced automatically as truth in simulation (dense and accurate) or they are approximated in the real-world (a gold standard).) Each of the different data attributes $A_{k}$ also has associated with it a list of $J_{k}$ number of fuzzy-predicates. For all experiments in this paper, we used the data attributes and fuzzy predicates in agreement shown in Table 2. We assumed that an object detector has been trained and evaluated on a set of test samples. For each of the data attributes, we have a set of fuzzy membership functions, $\mu_{k,j}\left(A_{k}\left(y_{n}\right)\right)$, which map the values associated with each of the *K* attributes to each of the fuzzy-predicates for each test sample.

### 2.1. Generating Statements

Our linguistic summaries are functionally equivalent to the form, “Q R Ys are P”, where Q is a quantifier, e.g., “Few”, “Some”, “Many”, Y is a data set, R is a qualifier, e.g., “Low confidence”, and P is a summarizer, e.g., “Low elevation angle, high altitude”. You may use Figure 4 to serve as an examples statement and its parts. To compute metrics on each summary, we first developed a scheme to enumerate and keep track of our linguistic statements. Let each statement, *S*, be represented by a fixed length sequence of integers, I(S). Each value in the sequence corresponds to a data attribute. If the value is 0, then that data attribute is not present in the current statement. Otherwise, the integer corresponds to an index in the list of fuzzy-predicates for that data attribute. For example, using the data attributes in Table 2, the sequence, [2, 1, 0, 0, 3, 0] corresponds to the linguistic statement, “Medium confidence, Low Altitude, and High Solar Elevation Angle images”. With this structure, we are able to enumerate all of the possible statements in a compact list of integers. Importantly, we are able to represent complex (e.g., [3, 2, 1, 2, 3, 2] = “High confidence, Medium Altitude, Low Camera Elevation Angle, Left Camera Azimuth Angle, Medium Solar Elevation Angle, Left Solar Azimuth Angle images”), highly specific statements just as easily as compact, simpler statements (e.g., [1, 0, 0, 0, 0, 0]→“Low confidence images”). Likewise, we can keep track of the quantifier and qualifier fuzzy-predicate index by appending two more values to the array. Using this framework, all of the possible statements are encoded as a 2D array of integers. The combination of all possible index values and thus statements can be efficiently generated by computing the Cartesian product of the set of index values for each attribute.

### 2.2. Linguistic Qualities

For each possible linguistic statement used to describe the performance of our detector, there are many different measures of quality we can compute. Depending on the use case, we may care more about statements which are highly accurate but are also relatively simple, or we may want statements that cover most of our data but are not very specific. Tuning the different criteria can help filter and guide results to be more useful. In this paper, we only considered four measures of quality: truth, focus, complexity, and operational relevance.

Probably the most popular measure is the “Truth Value”. This value represents how accurate a statement is with respect to a given dataset. The truth value, τ(S), for each statement is computed according to Equation (Equation 1). Statements, which are composed of data attributes and fuzzy-predicate pairs along with a quantifier, can be thought of as composing a fuzzy query on the dataset. Each sample, *n*, has an aggregate membership value for a given fuzzy query, which is the minimum across the fuzzy membership values to each of the data attributes and fuzzy-predicates defined by the statement, *S*. We use $\mu_{P}\left(y_{n}\right)$ to denote the membership of the *n*th sample, to the summarizer, *P*,
(1)$$\mu_{P}\left(y_{n}\right)=\min_{k,j\in I(s)}\mu_{k,j}\left(A_{k}\left(y_{n}\right)\right),$$
(2)$$\tau \left(Q\;R\;\mathrm{Ys}\;\mathrm{are}\;P\right)=\mu_{Q}\left(\frac{\sum_{n=1}^{N}\mu_{R}\left(y_{n}\right)\wedge \mu_{P}\left(y_{n}\right)}{\sum_{n=1}^{N}\mu_{P}\left(y_{n}\right)}\right).$$

Another common measure is called “Focus”. The focus value, sometimes called “coverage”, measures how much of the dataset supports a given statement. This can be useful when considering statements which are represented by very few samples. For example, consider the statement, “Many high confidence detections at low elevation angles” and a dataset where only 1 of the 10,000 images has a high membership value to “Low Elevation Angle”. If the image which has a high membership value to “Low Elevation Angle” also has a high membership value to “High Confidence”, then that statement will achieve a high truth value. This creates a problem because the statement explicitly states “Many ⋯ detections ⋯” even though only one image was used to compute the high truth value. Instead, we can also consider the “Focus”, weighing the fact that, in this example, the statement was only supported by a single image out of ten thousand. The focus value, F(S), for each statement is computed as,
(3)$$F\left(Q\;R\;\mathrm{Ys}\;\mathrm{are}\;P\right)=\frac{1}{N}\sum_{i=1}^{N}\mu_{Q}\left(\mu_{P}\left(y_{i}\right)\wedge \mu_{R}\left(y_{i}\right)\right).$$

As described above, the number of possible statements increases rapidly with the number of data attributes and quantities in agreement. In addition, the maximum length of a statement also increases, thought not as fast. Cognitive load refers to the amount of working memory required to perform a task [39]. Long statements composed of many attributes are likely to cause a higher cognitive load for a user to understand by requiring more working memory to process than shorter statements. Instead of reporting each and every place that the detector failed, we can summarize the performance in just one simple statement, “Many low confidence detections from the front”. As Kacprzyk stated, through linguistic summaries we hope to provide “intuitively appealing and comprehensible results to the human user” [40]. As such, one goal of linguistic summaries is to provide simple, accurate statements that demand a minimum cognitive load on the reader to understand. To measure the complexity of a summary, C(S), we chose to count the number of components via
(4)$$C\left(Q\;R\;\mathrm{Ys}\;\mathrm{are}\;P\right)=\frac{{∥I\left(S\right)∥}_{0}}{K},$$
where the numerator is the L0 norm, simply counting the number of attribute and fuzzy predicate pairs that are in a given statement. The denominator is the maximum number of attributes in a statement.

Finally, we considered a measure that functions mostly as a linguistic filter, which we call operational relevancy, O(S). For all possible statements of the linguistic summaries, we can define a weight which represents how important that statement is to us. For example, if we are mostly concerned with the failure modes, then we may want to place a relatively high weight on “Many” and “Low Confidences” so that statements describing failures are valued higher. Likewise, we could tune our results to focus mostly on medium and high altitudes by placing weights of [0.2, 1.0, 0.8, 0.1] for altitudes of “Low”, “Medium”, “High”, and “Very High”, respectively. To compute the relevancy for a given summary,
(5)$$w_{P}=\min_{k,j\in I(s)}w_{k,j},$$
(6)$$O\left(Q\;R\;\mathrm{Ys}\;\mathrm{are}\;P\right)=w_{Q}\wedge \frac{1}{N}\sum_{n=1}^{N}w_{R}\wedge w_{P},$$
where $w_{k,j}$ is the relevancy weight for the *j*th fuzzy-predicate for the *k*th attribute.

### 2.3. Aggregating Linguistic Qualities

To combine the different measures described above, there are many options. Ultimately, it is up to the user to define which characteristics are most important to them. We proposed one method of combining measures, with some flexibility to balance the measures as needed. For each summary, *S*, we have four measures: Truth = τ(S), Focus = F(S), Complexity = C(S), and Operational Relevancy = O(S). Before each of the measures we scale their values to be between 0 and 1 across all statements. We can then express relative simplicity as 1−C(S). To combine the measures, we used
(7)$$V\left(S\right)=\frac{O\left(S\right)\tau \left(S\right)[1+w_{F}F\left(S\right)+w_{C}\left(1-C\left(S\right)\right)]}{1+w_{F}+w_{C}},$$
and chose $w_{F}$ and $w_{C}$ to weight statements by how well supported by data or how simple the statements are. These variables tend to be highly correlated in terms of their outcome on resulting summaries. The statements are then ranked according to their final value, V(s), ordered greatest to least. A threshold could be applied to remove summaries which do not meet a minimum value. This aggregation strategy allows us to only present relevant, true statements while taking into account how supported by data and how simple each statement is.

## 3. Experiments and Results

### 3.1. Experimental Design, Model, and Dataset

In the preceding sections, a systematic process was outlined to generate linguistic summaries. Now, in this section, this method was applied to summarize the performance of an object detection model (In prior work [41], we showed how an earlier version of our approach can be applied to a publicly available object detection model). This model was trained using real data and is being used to evaluate simulated data with known truth. The training data for this model consisted of a relatively large number of labeled domain specific images captured by a drone at various low altitudes, depicting people at different stand-off distances and off-nadir slant angles. Overall, the resulting model is enigmatic, functioning as a black box. It takes a new image as input and produces predictions in the form of axis-aligned bounding boxes (AABBs), box centers, and associated confidences for the locations of people in the image.

While our primary emphasis lay in a person detector, we also ventured into exploring the training, testing, and evaluation of more unique and infrequent objects like explosive hazards (EH) [16,42,43,44] using both single and multiple modalities (RGB and infrared). It is essential to note that the specific object category, whether it is a person, EH, car, or other, is not the central focus of this article. Rather, our key objective was to showcase the process of gathering and evaluating an object-centric dataset acquired from a low altitude drone. To achieve this, we conducted simulations with different camera, platform, and environmental specifications. The drone equipped with a camera is maneuvered in a hemisphere pattern around the object, capturing data under diverse and varying conditions. This approach provides us with an ideal and controlled means to comprehensively study our AI/ML model. By manipulating and studying individual factors and their combinations, we can gain valuable insights into the model’s performance. Alternatively, if the reader can obtain accurate platform metadata and environment ground truth, real-world data can be used instead of simulated data.

To evaluate model performance, we created a simulated person dataset. We leveraged the Unreal Engine 5.1 and used content from the Epic Marketplace to construct a realistic scene depicting a person in a desert environment. Through the movie render queue, high-quality photorealistic images were obtained, covering a diverse range of conditions. Specifically, four distinct character models were used: light-skinned male, light-skinned female, dark-skinned male, and dark-skinned female. For each character, we generated imagery from eleven different camera azimuth angles, three camera elevation angles, six solar azimuth angles, three solar elevation angles, and ten standoff distances. This setup allows us to capture a wide array of scenarios for evaluation.

Figure 5 shows example imagery from our simulated person dataset. While we chose a relatively empty scene in an arid environment, primarily due to free high-quality photometrically scanned assets, we acknowledge the potential for a more diverse and robust dataset with varied scene conditions in the future. We selected this dataset to limit our analysis to a few variables and to minimize confounding factors. Our objective was a controlled and tractable experiment that can be used to study our linguistic summaries. The reader can clearly add additional objects, introduce occlusion, and other factors relative to their intended goal and application.

Figure 6 shows detector results, showcasing examples of the most confident hits, false alarms, and missed targets. For a more comprehensive understanding, Figure 7 shows the distribution of detection performance on our simulated dataset relative to our different data attributes. This plot serves the dual purpose of helping us validate the linguistic summaries and it acts as a summary in itself.

Figure 7 provides context for the following linguistic statements. The reader can see that the model has a peak average performance somewhere around a standoff distance of 53 m to the object (human). Performance decreases as distance increases, which could be expected as a function of the number of pixels on our target. However, performance drops off faster as the object is closer (thus larger and more pixels) to the camera/drone. This model behavior is more likely a function of the attributes of the labeled data versus a core deficiency in the object detection model. That is, there are little to no observations of this object at close distances to the drone. Therefore, we can derive that the model is sensitive to object scale. Next, Figure 7 indicates performance variation with respect to the relative camera azimuth angle, e.g., we see the human from their front, side, back, etc. Our training dataset had different human poses, but this variables distribution was not uniformly distributed. As such, performance variation could be an indicator of the data or difficulty in recognizing people from different poses. Further analysis would need to be sought to determine the culprit more deeply. Furthermore, solar elevation and azimuth angle create different performances, which changes the overall image intensity and shadows, but its effect looks negligible. Last, and perhaps most dramatically, camera elevation angle has a big impact, with the model performing better at lower angles and dropping off at/near Nadir (overhead). Again, these graphical summarizations of our model’s performance relative to our camera, platform, and environment attributes will help set the stage for understanding our linguistic statements.

### 3.2. Tailored Summaries

To showcase the flexibility and adaptability of our approach in accommodating individual user preferences, we produced distinct sets of results tailored for two specific users. This demonstration exemplifies how summaries can be customized to meet the unique needs and criteria of different users. By presenting diverse summaries for each user, we underscored the versatility of our method in delivering relevant and personalized information to various stakeholders.

“User A” embodies the role of a research scientist, assigned with the crucial task of comprehending the existing model’s limitations and devising strategies for enhancement. For this user, the primary emphasis lies in scrutinizing the failure cases characterized by numerous low confidence statements. They are entirely at ease with receiving lengthy, detailed statements ($w_{C}=0$), provided they are precise and accurate. To optimize each improvement iteration, User A prefers to concentrate solely on the most valid statements, showing little concern for focus and complexity ($w_{F}=0$ as well). Their objective is to extract the most relevant and truthful information from the summaries, enabling them to efficiently prioritize and address the model’s weaknesses in subsequent iterations.

“User B” serves as a non-technical marketing designer, responsible for presenting the most favorable aspects of the current model to stakeholders. Their key focus lies in success cases distinguished by numerous high confidence statements. As a marketing professional, User B prefers simple and easily understandable statements (high $w_{C}$), prioritizing clarity and accessibility in the summaries. In this context, User B is willing to accept a trade-off where some true and operationally relevant statements may not make up a substantial portion of the dataset (low $w_{F}$). Their objective is to showcase the model’s strengths in the most user-friendly manner, highlighting the positive outcomes and confidently conveying the model’s capabilities to stakeholders.

### 3.3. Results

To offer in-depth insights into the evaluations tailored for each user, we present the results of running the linguistic summary separately for User A and User B in Table 3 and Table 4, respectively. These tables provide a comprehensive breakdown of the linguistic summaries specifically generated for each user, precisely catering to their individual preferences and objectives. By presenting user-specific outcomes side by side, we demonstrate the customized nature of our approach, showcasing how linguistic summaries can be fine-tuned to suit the unique needs of different stakeholders.

The summary for User A was tuned to focus on “Many low confidence ⋯” statements. This user wanted statements that are accurate and relevant to them, even if they were very detailed and specific. As a result, the summary produces statements that on average have a simplicity value of only 0.1, but a very high minimum truth value of a perfect 1.0. By analyzing the top ranked statement, “Many low confidence close standoff distances front right object azimuth angle high sun elevation angle front right sun azimuth angle detections” against the violin plots and the membership functions in Figure 3, one can confirm that this is an accurate statement, given the information our summarizer had.

The summary for User B was tuned to focus on “Many high confidence ⋯” statements. This user was interested in an accurate and relevant statements as well. However, this user very much preferred compact summaries supported by a relatively large proportion of the data. This resulted in an average simplicity score of 0.64. However, the minimum truth value was now 0.514 for the lowest ranked statement. The top ranked statement, “Many high confidence far standoff distances low camera elevation angle detections” is accurate. If we look back at the plots, we see that, at far standoff distances, the detector preforms relatively poorly at far standoff distances and there is a large amount of low confidence low camera elevation angle detections as well. Both summaries are valid but differ in important ways based on the user inputted criteria.

An intriguing perspective on the structure of the statements generated by our method is to envision the summary hierarchically. Beginning with the simplest statements and progressively building up to more complex ones, a directed graph can be constructed to represent this hierarchy. It is important to recall that each summarizer is composed of attribute and fuzzy predicate pairs. In our constructed graph, edges connect nodes containing the current node’s string as a sub-string, with only one additional attribute. For example, “Many high confidence detections” would be connected to “Many high confidence detections at medium standoff distances”. This connection scheme ensures that the graph captures the relationships between statements as they become increasingly elaborate. Visualizing this topology in two dimensions is challenging due to the richness and intricacy of the linguistic summaries. As we delve deeper into the hierarchical tree, creating more complex statements, we discover new connections between branches that were not previously evident. This exploration encompasses all possible statements and connections based on their complexity, resulting in a complex and multi-dimensional structure. To help navigate and comprehend this linguistic summary graph, we utilized the Gephi tool, which offers valuable visualization capabilities [45]. See Figure 8 for a visualization of the linguistic summaries for User A. Nevertheless, we are actively exploring alternative methods to construct and visualize the graph in a manner that maximizes its utility and facilitates a better understanding of the summarization process.

### 3.4. LLM Summaries of Lingustic Protoform Summaries

In recent times, large language models (LLMs) like ChatGPT [46] have gained unparalleled popularity due to their versatility in various applications, including summarization. However, the linguistic summaries produced by our method might still contain redundant phrases. Combining several statements into a simpler yet equally accurate summary could be beneficial. An existing challenge with LLMs is their sensitivity to the exact “prompt” or input text, as evident in studies such as [47,48]. As a quick demonstration of leveraging an LLM to enhance summaries, we fed our statements into ChatGPT using GPT-4 as the base model. Below are the results obtained via the prompt: “The following statements were produced by a linguistic summarizer of an aerial object detection model. Can you please analyze them and reduce any redundancy to produce a more succinct report? Can you summarize it in one sentence while maintaining as much of the content as you can?”

User A: *The aerial object detection model identified numerous low confidence detections at close standoff distances, with varied camera elevations, object azimuths, and sun angles, primarily focusing on medium camera elevation and right object azimuth scenarios.*

User B: *The aerial object detection model demonstrates high confidence in detecting objects at far standoff distances with low and medium camera elevation angles, as well as various object azimuth angles, including front left, back, and left.*

This simple experiment merely scratches the surface of the vast possibilities that arise from harnessing LLMs for model summarization. The summaries produced by ChatGPT are remarkably accurate and impressive, especially considering the relatively unstructured input data of natural language followed by machine-produced results. Imagine the potential if we were to run an LLM like ChatGPT offline, enabling us to automate the generation process using highly tailored prompts that incorporate a broader range of input data beyond the ten statements provided here. This approach presents an appealing prospect as it combines the strengths of principled approaches, yielding true and relevant statements, with the added benefit of simple, well-articulated, and user-friendly results, making it a compelling choice for effective summarization.

## 4. Conclusions and Future Work

In conclusion, this article presented a novel Explainable AI (XAI) approach designed to generate linguistic summarizations of black box models, offering tailored explanations to suit diverse user needs. We applied this approach in the context of explaining an object detector model, e.g., automatic target recognition (ATR), utilized in a low-altitude aerial drone setting. Two very different types of users were demonstrated and example summaries were provided. Moreover, we delved into the potential of large language models (LLM) to further refine and simplify our summaries for human consumption. To demonstrate the efficacy, capabilities, and flexibility of our approach, we conducted an experiment using simulation with known ground truth. Through this experiment, we successfully showcased how our methodology empowers users to gain insights into the inner workings of an AI/ML model and its decision-making process. By producing comprehensive and accountable user-specific linguistic summaries, we have taken a step towards bridging the gap between complex black box models and human interpretable explanations.

With respect to future work, we are keen on further investigating the impact of the underlying evaluation dataset’s distribution on our generated linguistic summaries. It is crucial to approach the dataset selection meticulously, ensuring that the attributes are uniformly distributed throughout the dataset to prevent any artificial biases in the summaries. To address this challenge, we propose two potential solutions. Firstly, one can thoughtfully curate the evaluation dataset, as we have done in our study, ensuring that it encompasses a well-balanced representation of attributes to foster unbiased summaries. Alternatively, we can adapt our technique to handle knowledge gaps or biases that may exist in the distribution of the evaluation dataset. This adaptation would enable us to produce reliable and accurate summaries even under varying dataset conditions.

Moreover, we are interested in exploring alternative aggregation operators and linguistic qualities in pursuit of user multi criteria decision making. By expanding our range of options, we can exercise greater control over the characteristics of the resulting summaries, tailoring them to specific user requirements more effectively.

The graph-based hierarchical structure and the corresponding visualizations have proven invaluable in understanding the data in ways that text summaries alone cannot achieve. These visualizations offer a more intuitive and comprehensive grasp of the summarization process, providing valuable insights into the complex relationships between statements, which may otherwise remain obscured in textual form. Future work looking into interactive processes or combined graphical and text explanations is important.

A highly promising use case that our method opens up for exploration is its integration into a closed-loop machine learning and data generation system. By feeding our linguistic summaries into procedural generation tools, we can generate photorealistic imagery, particularly in scenarios where our model’s performance is subpar. This approach allows us to delve into the inherent difficulty of certain scenarios and gain a deeper understanding of their impact on object detection. For instance, scenarios with less contrast between the target and the background are expected to pose increased challenges in detection, irrespective of the training data. By exploring such scenarios, we can shed light on the underlying factors affecting our model’s performance and develop strategies to overcome these challenges. Our ultimate goal is to develop an automated system capable of achieving exceptional object detection proficiency across diverse scenarios. This system would be self-improving, continuously learning and adapting to changing operating conditions. By combining our linguistic summarization approach with procedural data generation and machine learning, we aim to create an intelligent and agile system that can dynamically enhance its capabilities, staying at the forefront of object detection performance.

Last, we employed simulation to acquire imagery and precise metadata. While our process and results hold valid within this context, we recognize the significance of investigating the impact of metadata errors on the summary generation process. Ensuring robustness in the presence of potential inaccuracies in metadata is an important avenue for further exploration. While our use of fuzzy set theory naturally absorbs some level of error, a precise study and possible extensions might be warranted. Additionally, to maintain accuracy and minimize confounding factors, we deliberately restricted the complexity of our environment. This approach allowed us to focus on producing accurate summaries without overwhelming complexities. However, for future work, it will be essential to broaden our scope and include a more diverse set of variables. By introducing more complex scenes, we can gain a deeper understanding of how our method performs under varied and challenging conditions. This expansion will enable us to assess the generalizability and adaptability of our approach across a wider range of scenarios and applications.

## Figures and Tables

**Figure 1 sensors-23-06879-f001:**
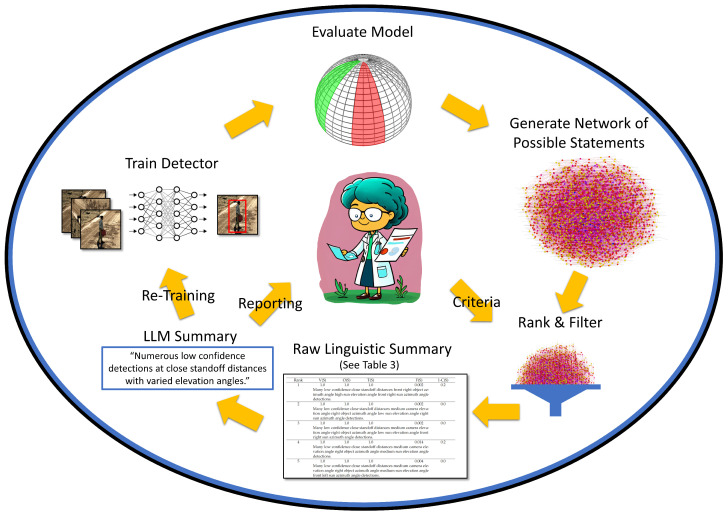
Overview of the process for generating and using linguistic summaries. First a detector is trained, then evaluated. The red box indicates an object declaration from a trained detector. A network of possible statements describing the model is generated and filtered based on user specific criteria. Linguistic qualities are computed and either a final report is given to the user or these statements are used to generate new data and train a new improved model.

**Figure 2 sensors-23-06879-f002:**
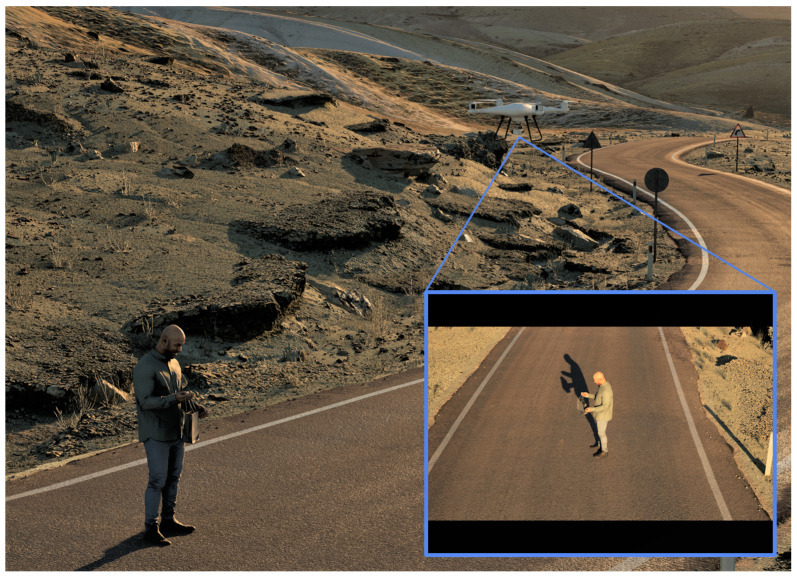
Example generated in Unreal Engine 5 of a virtual drone collecting imagery for a target object (person). This image illustrates one instance of collecting data in a hemisphere pattern at different relative poses, standoff distances, and environmental conditions (time of day, fog, etc.). The result is a virtual dataset (complete with ground truth) that is passed to a black-box deep learning object detector trained using real-world data. Finally, the detectors output is passed to our process for generating linguistic summaries of performance.

**Figure 3 sensors-23-06879-f003:**
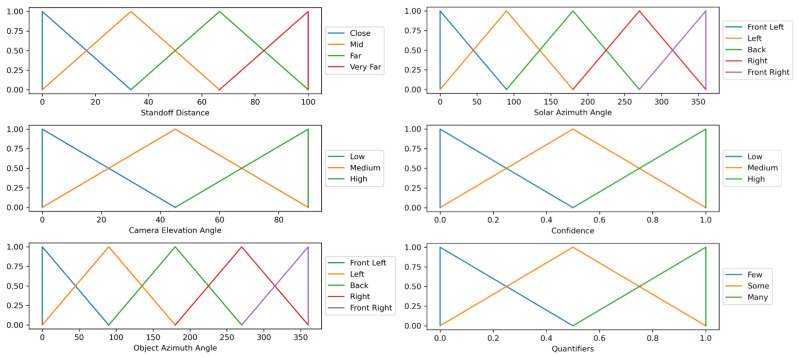
Plot of the membership functions used to calculate the quality measures and final ranking for each linguistic statement. Each attribute has a number of predicates. For each attribute + predicate pair, there is a membership function that can be tailored to match heuristic knowledge. Triangular functions were used to create a more decisive model. While we chose these functions for this paper, a user can choose whichever functions they desire.

**Figure 4 sensors-23-06879-f004:**
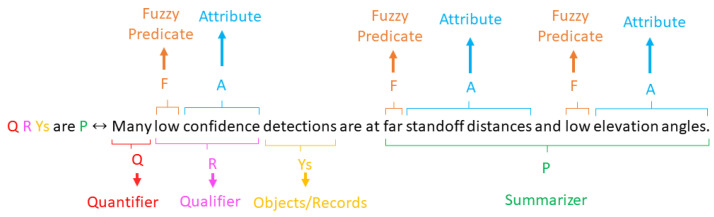
The parts breakdown of a linguistic summary of the form “Q R Ys are P”. Q represents a quantifier. R represents a qualifier. Ys are the data samples in the data set. P is a summarizer. Summarizers and qualifiers are made up of attribute + fuzzy predicate pairs.

**Figure 5 sensors-23-06879-f005:**
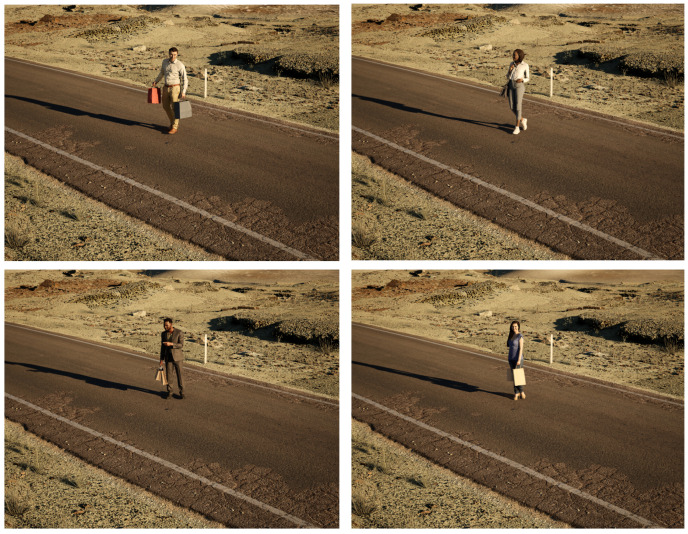
Example renders for each of the different character models used in our simulated person dataset. Each model was placed in the same location in an arid landscape environment. Images were captured using the Movie Render Queue in Unreal Engine under a variety of different conditions. Each data attribute that we vary such as standoff distance, camera angle, and solar angle, are recorded for each sample. They are directly used to produce the linguistic summaries.

**Figure 6 sensors-23-06879-f006:**
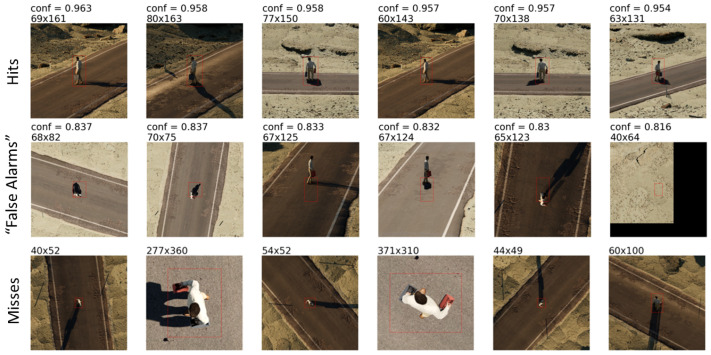
Plots visualizing the detection results of the object detector on the light-skinned male subset of our simulated person dataset. The confidence is given for the “Hits” and “False Alarms”, and the bounding box size (width × height) is given for each declaration shown. (**Top**): Highest confidence true positive detections. (**Middle**): Highest confidence false alarms. Declarations over a target that are too large were scored as incorrect, as well as declarations that did not meet a minimum IOU (intersection over union) threshold. (**Bottom**): Examples of missed targets, randomly shuffled.

**Figure 7 sensors-23-06879-f007:**
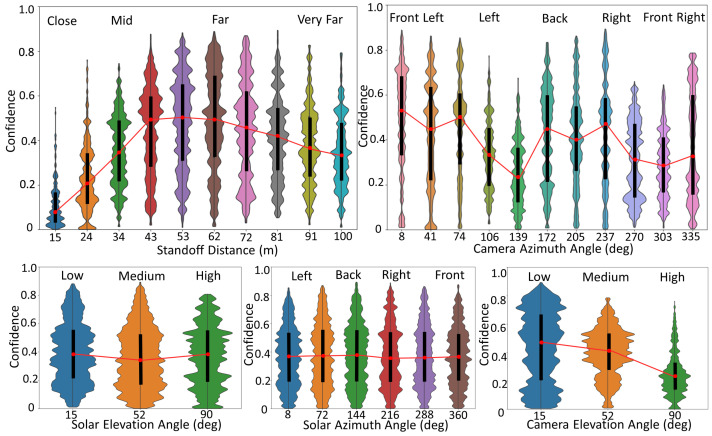
Plots visualizing the performance of the object detector as a factor of the different attributes. The width of each of the “violins” is determined by the number of occurrences of a particular attribute (*x*-axis) at a particular confidence. The thick black line in the middle of each unique data attribute value represents the interquartile range. The small red dot in the center of each thick black line represents the median. Although it is labeled only as “Confidence” for simplicity, the *y*-axis is actually IOU * confidence. These visualizations of performance are dictated by scored results, not just raw detector confidence. (**Top Left**): Standoff distance. (**Top Right**): Camera azimuth angle. (**Bottom Left**): Solar elevation angle. (**Bottom Middle**): Solar azimuth angle (**Bottom Right**): Camera elevation angle.

**Figure 8 sensors-23-06879-f008:**
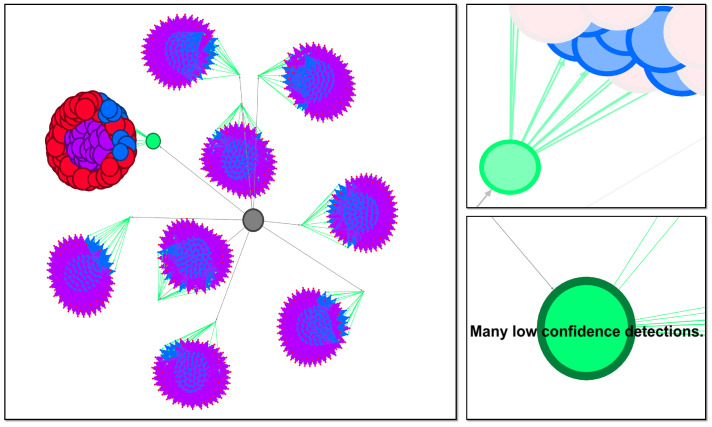
Graphic visualization of linguistic summaries for User A. Each node is a statement, sized by the final value assigned to that statement. Each edge is a directed connection from a simple statement to a more complex statement. The color of each node and edge is determined by the complexity of the statement. Each unique complexity value gets a random, unique color. (**Left**): Overview of all of the possible statements. Most of the statements have a zero value (no operational relevancy for User A, so they are set to the minimum node size. The “Many low confidence ⋯” branch is the only one with non zero values. (**Top Right**): Zoomed in view of a few nodes and their edges for the relevant branch of this summary. The green node is selected so only its immediately connected nodes are highlighted. (**Bottom Right**): Zoomed in view of a single node, its connections, and its corresponding statement.

**Table 1 sensors-23-06879-t001:** List of notation used in this paper.

Notation	Description
*Q*	Quantifier (e.g., few, some, many)
*R*	Qualifier (fuzzy predicate + attribute)
$y_{n}$	*n*th data sample
*P*	Summarizer
$A_{k}$	*k*th data attribute (e.g., altitude)
$F_{k,j}$	*j*th fuzzy predicate for the *k*th data attribute
μ	fuzzy membership function
*S*	statement (Q R Ys are P)

**Table 2 sensors-23-06879-t002:** Data attribute and predicate pairs considered in this paper.

Data Attribute ($A_{k}$)	$J_{k}$	Fuzzy Predicates ($F_{k,j}$)
Altitude	4	Low, Medium, High, Very High
Camera Elevation Angle	3	Low, Medium, High
Camera Azimuth Angle	5	Back, Left, Front Left, Front, Right, Front Right
Solar Elevation Angle	3	Low, Medium, High
Solar Azimuth Angle	5	Back, Left, Front Left, Front, Right, Front Right
Confidence	3	Low, Medium, High

**Table 3 sensors-23-06879-t003:** Linguistic summary for User A. This user is focused primarily on where the model fails (many low confidence). This user is mostly concerned with statements that are true and relevant, with $w_{c}=0$ and $w_{F}=0$. They are comfortable with long, detailed statements. This summary contains the 10 statements with the highest aggregate score.

Rank	V(S)	O(S)	T(S)	F(S)	1 − C(S)
1	1.0	1.0	1.0	0.002	0.2
	Many low confidence close standoff distances front right object azimuth angle high sun elevation angle front right sun azimuth angle detections.				
2	1.0	1.0	1.0	0.002	0.0
	Many low confidence close standoff distances medium camera elevation angle right object azimuth angle low sun elevation angle right sun azimuth angle detections.				
3	1.0	1.0	1.0	0.002	0.0
	Many low confidence close standoff distances medium camera elevation angle right object azimuth angle low sun elevation angle front right sun azimuth angle detections.				
4	1.0	1.0	1.0	0.014	0.2
	Many low confidence close standoff distances medium camera elevation angle right object azimuth angle medium sun elevation angle detections.				
5	1.0	1.0	1.0	0.004	0.0
	Many low confidence close standoff distances medium camera elevation angle right object azimuth angle medium sun elevation angle front left sun azimuth angle detections.				
6	1.0	1.0	1.0	0.005	0.0
	Many low confidence close standoff distances medium camera elevation angle right object azimuth angle medium sun elevation angle left sun azimuth angle detections.				
7	1.0	1.0	1.0	0.005	0.0
	Many low confidence close standoff distances medium camera elevation angle right object azimuth angle medium sun elevation angle back sun azimuth angle detections.				
8	1.0	1.0	1.0	0.005	0.0
	Many low confidence close standoff distances medium camera elevation angle right object azimuth angle medium sun elevation angle right sun azimuth angle detections.				
9	1.0	1.0	1.0	0.004	0.0
	Many low confidence close standoff distances medium camera elevation angle right object azimuth angle medium sun elevation angle front right sun azimuth angle detections.				
10	1.0	1.0	1.0	0.002	0.0
	Many low confidence close standoff distances medium camera elevation angle right object azimuth angle low sun elevation angle back sun azimuth angle detections.				

**Table 4 sensors-23-06879-t004:** Linguistic summary for User B. This user is focused primarily on where the model succeeds (many high confidence). This user is mostly concerned with simple statements that are true, relevant, and make up a significant portion of the data with $w_{c}=1$ and $w_{F}=1$. This summary contains the 10 statements with the highest aggregate score.

Rank	V(S)	O(S)	T(S)	F(S)	1 − C(S)
1	0.34	1.0	0.782	0.08	0.6
	Many high confidence far standoff distances low camera elevation angle detections.				
2	0.334	1.0	0.732	0.131	0.6
	Many high confidence far standoff distances medium camera elevation angle detections.				
3	0.312	1.0	0.756	0.029	0.6
	Many high confidence low camera elevation angle front left object azimuth angle detections.				
4	0.311	1.0	0.576	0.128	0.8
	Many high confidence low camera elevation angle detections.				
5	0.307	1.0	0.952	0.018	0.4
	Many high confidence far standoff distances low camera elevation angle front left object azimuth angle detections.				
6	0.299	1.0	0.522	0.203	0.8
	Many high confidence medium camera elevation angle detections.				
7	0.294	1.0	0.701	0.047	0.6
	Many high confidence medium camera elevation angle front left object azimuth angle detections.				
8	0.292	1.0	0.674	0.078	0.6
	Many high confidence medium camera elevation angle back object azimuth angle detections.				
9	0.291	1.0	0.67	0.079	0.6
	Many high confidence medium camera elevation angle left object azimuth angle detections.				
10	0.289	1.0	0.514	0.178	0.8
	Many high confidence far standoff distances detections.				

## Data Availability

Data sharing not applicable.

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
