# Peer review of "Linguistic Explanations of Black Box Deep Learning Detectors on Simulated Aerial Drone Imagery"

_sensors, 2023, doi:10.3390/s23156879_

Round 1
Reviewer 1 Report
This paper proposed a hierarchical approach for generating and evaluating linguistic statements of black box deep learning detectors, which is very attractive in a closed loop machine learning and data generation system for aerial image analysis.
The overall structure of the paper is clear and the content is substantial, but there are still some contents that need to be further improved. The following are my comments:
(1) In the introduction part, the introduction of relevant work on large language models (LLMs) is lacking, and it is suggested to supplement. At the same time, the motivation can be further improved, and the writing can be more clear and prominent
(2) On line 126. Section 2.1, experiments and results are provided. ‘Section 2.1’ -- ‘Section 3’?
(3) On lines 132-134, as the authors explain, each of the different data attributes Ak, also has associated with it a list of Jk number of fuzzy-predicates. However, as shown in Table 2, the Camera Azimuth Angle and Solar Azimuth Angle, the Jk may be 5 according to the Fuzzy Predicates, why 4?
(4) On line 198, no space is needed, it is simply an explanation of Equation 4. The same goes for line 209.
(5) In the experiment section and results section, it is recommended to create subheadings (such as Dataset and Experimental Setup, Evaluation Method, Results) to make it easier and clearer for readers to understand the setup and effects of the experiment.
(6) Is there any qualitative or quantitative metric to evaluate the accuracy and trustworthiness of a method for generating linguistic summaries proposed by the author?
Reviewer 2 Report
Introduction should include a wider overview of alternative solutions already published in relevant literature.
line 34: to include a reference for Unreal Engine
line 52, 61, 102, 122, 142,
line 97: Yager first introduced ..... Include a reference.
line 189: Lengthier statements.... Include a reference.
line 198: when you use "where", do not use margin and include a comma after the term "where".
line 267-269: examples with different, more complex scenes, should be included.
line 292: use the term "Conclusions"
The paper needs to include real cases on images obtained by real drones. In this way noise and additional disturbances could be included.
Avoid the foot notes.
Avoid the use of terms "we", "our"
Reviewer 3 Report
Major revisions on the presentation of figures to improve their quality, table titles to be placed on top of tables, re-structuring of sections 3 and 4, and adding an additional review to strengthen findings.
Tables 1 and 2: Table titles are usually above the table as the figure captions are placed below the figures.
Section 2 may state "Theoretical Background"
Section 3 (the first part of it) may state "Methodology or Materials and Methods"
Line 277: Tables 3 and 4 show......
Figure 7: The text on x and y axes is not readable, improve its presentation and the presentation of other such figures.
Figure 8 (bottom right): the text is not readable.
Section 4 (newly proposed for Results and Discussion) needs further discussion on results. This will also increase references used in this manuscript, which restrict to 30 only as of now.
Can be improved as commented,
Figure 3 and 4 shows must be changed to Figures 3 and 4 show .....
Round 2
Reviewer 1 Report
This paper explore a new and novel process to automatically generate a summarized set of linguistic statements of a black box AI/ML model. After the revision, the quality of the article has been improved to some extent, but the overall chapter structure arrangement still needs more thinking. Here are some of my comments:
1. Some fonts in Figure 7 are too small. It is suggested to increase the clarity of images 1-2.
2. There is coupling between the methodology of Section 3 and the results of Chapter 4. The overall structure does not feel smooth.
3. In general, the conclusion should be more concise. The discussion chapter can be added to analyze the relevant experimental results and discuss the future research direction.
Author Response
"Please see the attachment.

Reviewer 2 Report
My recommendations were not included in the revised manuscript.
do not use the terms "we", "our".
Author Response
No new input for Reviewer 2. Edits were made in last revision.
Reviewer 3 Report
The conclusion section is too long. The readers need to read just of the work in this section. Suggest moving some of the text, tables, and the relevant discussion to the Results section and name it “Results and Discussion” instead of Results only. No other issues found. Good luck.
